# The Relationship between Sources of COVID-19 Vaccine Information and Willingness to Be Vaccinated: An Internet-Based Cross-Sectional Study in Japan

**DOI:** 10.3390/vaccines10071041

**Published:** 2022-06-29

**Authors:** Takeshi Yoda, Benjamas Suksatit, Masaaki Tokuda, Hironobu Katsuyama

**Affiliations:** 1Department of Public Health, Kawasaki Medical School, Kurashiki 701-0192, Japan; katsu@med.kawasaki-m.ac.jp; 2Department of Health and Sports Science, Kawasaki University of Medical Welfare, Kurashiki 701-0193, Japan; 3Faculty of Nursing, Chiang Mai University, Chiang Mai 50200, Thailand; benjamas.s@cmu.ac.th; 4Center for International Research and Cooperation, Kagawa University, Takamatsu 760-0016, Japan; vp-tokuda@kagawa-u.ac.jp

**Keywords:** vaccine hesitancy, vaccine refusal, information sources, COVID-19, misinformation

## Abstract

Despite considerable interest in the Japanese population in receiving the vaccine for COVID-19 when it first became available, a sizable percentage of people remain unwilling or hesitant to be vaccinated. Concerns among both the vaccinated and the unwilling center on the vaccine’s efficacy and its safety. Thus, this study aimed to identify whether the willingness to receive COVID-19 vaccination is related to the sources of information people use to learn about the vaccine. A cross-sectional study was conducted on 800 participants registered in an Internet research panel across Japan who completed a questionnaire on their sources of information about the vaccine, demographics, and vaccination status. Vaccine willingness/hesitancy and refusal were set as dependent variables in the logistic regression analysis, with sources of vaccine information and other socio-demographic variables set as independent variables. The results of the analysis found that the information sources significantly associated with willingness to vaccinate were TV (AOR 2.44 vs. vaccine refusal/hesitation), summary websites of COVID-19 by non-experts (AOR 0.21, vs. vaccine refusal/hesitation), Internet video sites (AOR 0.33, vs. vaccine refusal/hesitation), and the personal websites of doctors (AOR 0.16, vs. vaccine refusal/hesitation). Given the likelihood of misinformation in non-traditional sources of information, it is important that health communications be accurate and persuasive.

## 1. Introduction

Since first diagnosed in December 2019 in Wuhan, Hubei Province, China, novel coronavirus infection (COVID-19) became recognized as a global threat infection a few months later [1]; as of April 2022, the pathogen, SARS-CoV-2, has six variants and has infected approximately 495 million people, of whom 6.16 million have died [2]. To combat this unknown infectious disease, vaccines and therapeutics have been developed around the world. With regard to vaccines, more than 60 candidates against COVID-19 had been investigated globally only four months after the novel pathogen was reported [3]. In particular, three vaccines had ≥90% efficacy rates: the BNT162b2 vaccine (95% protection against COVID-19) [4], the ChAdOx1 nCoV-19 vaccine (90% protection against COVID-19) [5], and the mRNA-1273 vaccine (94.1% protection against COVID-19) [6]. In the United States in December 2020, the Food and Drug Administration issued an emergency use permit for Pfizer–Biontec’s BNT162b2 vaccine, and vaccination began at the end of 2020. The World Health Organization (WHO) has now certified six different COVID-19 vaccines and is calling for people around the world to be vaccinated.

Immunization in Japan began in February 2021, a few months after the start of immunization in the United States. Vaccination coverage was first extended to healthcare workers, then to older people and people with underlying medical conditions at the end of March, and soon after to all of the adult population [7]. We found that as of September 2020, approximately 65% of the population wanted to be vaccinated [8], and the number of those who wanted to be vaccinated increased as the vaccination program began; by April 2022, nearly 80% of the population had received two doses [9]. In the vaccination willingness survey undertaken during the development phase, both those who wanted to be vaccinated and those who did not or who were hesitant showed very high interest in the “safety” and “efficacy” of the vaccine. The SAGE Working Group on Vaccine Hesitancy in the UK concluded that factors of vaccine hesitancy could be grouped into three categories: contextual, individual, and group and vaccine/vaccination-specific influences [10]. According to previous studies, the most important factor in the hesitation to receive the COVID-19 vaccine, besides general vaccine avoidance, appears to be distrust of the safety of the vaccine (including uncertainty due to the novelty of the vaccine) [11,12]. In the case of the COVID-19 vaccine, it took only one year from the discovery of the pathogen to the development and commercialization of the vaccine. Traditionally, it has taken years, even decades, to develop vaccines, from about 40 years for polio to 5 years for Ebola, with most vaccines taking an average of 15 years. [13,14]. Moreover, the COVID-19 vaccine is a new way of making vaccines, using messenger RNA (mRNA). Conventional vaccines are inactivated viral proteins or viruses themselves, which stimulate the body’s immune defenses when infected with a live virus. However, the first two COVID-19 vaccines whose efficacy was announced in a large clinical trial (Phase III) were simply strings of mRNA in a lipid coat [15]. Therefore, it was not surprising that some groups questioned the safety of the COVID-19 vaccine and the existence of COVID-19 itself. Moreover, the spread of scientifically incorrect information through social networking services and other means has been addressed in various news reports and articles [16,17,18,19]. The fact that a certain number of people believe this biased information and refuse or are hesitant to receive the vaccine has become a worldwide problem.

In Japan, this phenomenon is no exception, and currently, there are still some who are vaccine-hesitant or refusers (approximately 10% from our former study and others). In particular, the vaccination rate in children has been sluggish due to a lack of information and excessive fear of adverse reactions [20].

## 2. Materials and Methods

We conducted this study in Japan in December 2021 and employed the Internet research panel data from QiQUMO, which is operated by Cross Marketing Inc., Tokyo, Japan. More than two million people were registered in this research panel in Japan. The sample size was calculated using the following formula:

n = λ^2^*p*(1 − *p*)/d^2^,

where n is the sample size, λ is the confidence level, *p* is the response ratio, and d is the tolerance. We substituted λ = 1.96, *p* = 0.5, d = 0.035, resulting in a minimum sample size of 784. We allowed a larger margin to account for incomplete responses and identified 800 respondents in total.

The questionnaire consisted of eight questions seeking the following information:

(1) Sex, (2) age, (3) occupation, (4) educational background, (5) presence of chronic diseases (excluding COVID-19 infection), (6) willingness to receive the COVID-19 vaccine (derived from the questions: “Would you be willing to be vaccinated against COVID-19?” or “Have you already received the COVID-19 vaccination?” with the response options being “yes,” “unsure,” and “no”), and (7) information sources for COVID-19 vaccines (multiple answers). These items were adopted from Japanese and Thai studies [8,20,21]. The sources of information for COVID-19 vaccines were as follows: (1) television (TV) news, (2) newspapers, (3) weekly magazines, (4) websites of the Ministry of Health, Labour and Welfare (MHLW)/National Institute of Infectious Diseases (NIID), Japan, (5) Covi-Navi website, a collection of correct COVID-19 information by Japanese infectious disease specialists, (6) medical associations’ and public health centers’ websites, (7) pharmaceutical companies’ websites, (8) summary websites of COVID-19 information created by non-experts, (9) Social Networking Services (SNS) such as Facebook and Twitter, (10) Internet video sites such as YouTube and TikTok, (11) doctors’ personal websites, (12) non-doctors’ personal websites, (13) family doctors, (14) neighborhoods/friends/families, or (15) publicity or direct visits from local government offices/health centers.

Descriptive statistics were used to evaluate vaccine willingness by sex, age group, occupation, educational background, and presence of chronic diseases. The relationship between COVID-19 vaccine information source and participants’ willingness or hesitancy/refusal to receive the vaccine was expressed as a percentage of each item since multiple responses were provided. The chi-squared test was applied to evaluate the categorical variables. We also analyzed the characteristics of the relationship between vaccine willingness, sources of information, and other socio-demographic variables using logistic regression analysis. For logistic regression analysis, vaccine willingness/hesitancy and refusal were set as the dependent variables, with sources of COVID-19 vaccine information, sex, age group, educational background, and presence of chronic diseases set as the independent variables. The significance level was set at <0.05. JMP Pro 14.1.0 (SAS Institute Inc., Cary, NC, USA) was used for all of the analyses.

The study was approved by the ethical committee of the Kawasaki Medical School (approval number: 5499-00). Implied consent rather than formal written consent was sought to ensure the anonymity of the participants. The participants clicked the “I agree” button before commencing the survey to indicate their consent.

## 3. Results

Of the 800 participants, 418 were males (52.2%), and the average age was 44.6 years old (standard deviation was 20.3). Other socio-demographic characteristics are presented in Table 1.

Overall, 640 (80.0%) out of 800 participants stated that they were either willing to be vaccinated or were already vaccinated against COVID-19. A further 84 (10.5%) participants replied that they were unsure, and 76 (9.5%) stated that they did not intend to get vaccinated. As illustrated in Table 2, we found considerable differences in willingness to get vaccinated by sex, presence of chronic diseases, educational background, and occupation (Table 2).

Figure 1 shows a bar chart of the sources of information for the COVID-19 vaccine for each of the respondents by whether they wanted to be or had been vaccinated or did not want to be and were hesitant to be vaccinated. The two groups (those who did not want to be vaccinated or were hesitant) were combined due to the small number of respondents. The vaccine willingness/already vaccinated group was more likely than the other groups to use TV and newspapers as sources of information. On the other hand, those who refused or hesitated to be vaccinated were more likely to use SNS, Internet video sites, and personal websites by doctors as sources of COVID-19 vaccine information.

To determine the relationship between sources of information and willingness to be immunized, we conducted a logistic regression analysis with the willingness to be vaccinated as the dependent variable. Each model was adjusted as follows: Model 1 used only sources of COVID-19 vaccine information as the independent variable; Model 2 added sex and age group as independent variables in addition to Model 1; Model 3 further adjusted for annual income group, educational background, and presence of chronic diseases as independent variables. Occupation was not used as an independent variable because of the occurrence of categories with zero items. In the model adjusted for other factors, the information sources significantly associated with willingness-to-vaccinate were TV (AOR 2.44 vs. vaccine refusal/hesitation), summary websites of COVID-19 by non-experts (AOR 0.21, vs. vaccine refusal/hesitation), Internet video sites (AOR 0.33, vs. vaccine refusal/hesitation), and personal websites (blogs) of doctors (AOR 0.16, vs. vaccine refusal/hesitation) (Table 3).

## 4. Discussion

In our survey, 80% of the respondents indicated that they wanted to or had already received the COVID-19 vaccine. This is comparable to or slightly higher than the previously reported willingness-to-vaccinate survey of the Japanese population [8,22,23]. When compared to the vaccination rates (80.5% had received at least two doses of vaccine) published by the Prime Minister’s office, the figures in our study are almost equal to the rates of the Japanese population, as 20% of the participants refused vaccination or were hesitant.

From the results of the descriptive statistics shown in Table 2, the characteristics of those hesitant/refusing to be vaccinated are as follows: women, younger generation (under 30 years old), no reports of specified chronic conditions, and lower educational level. These results are consistent with those of previous studies. Khan et al. reported that a Japanese COVID-19 vaccination awareness survey showed almost similar results [24]. While surveys in several other countries have reported that women are less hesitant or less likely to refuse to be vaccinated than men [25], Japan has historically reported more mothers who are hesitant to vaccinate their children [26,27], and other studies have found women to be more hesitant than men [20,28]. In addition, the highest annual income group (more than JPY 12 million) seemed to have a higher percentage of vaccine hesitancy/refusal than the others, but the difference did not reach statistical significance. Further numbers were needed, as this was due to a too small number of people in the category. Furthermore, the results of the logistic regression analysis, which was adjusted for other factors, shows the characteristics of the vaccine-hesitant/refusal group are more clearly defined as follows: (1) younger generation (Based on age 19 or younger. In Model 2, each age group over 40 had significantly higher odds ratios across the board.), (2) those with relatively lower incomes (In Model 3, the odds ratio of JPY 3–6 million was significantly lower), (3) those whose sources of vaccine information include private websites (In Model 3, the odds ratios were significantly lower for the items summary websites of COVID-19 by non-experts, Internet video sites, and doctors’ personal websites.) In one study, it was reported that COVID-19 vaccine refusal and vaccine hesitancy were both significantly associated with females, age (in an inverted U-shaped relationship), lower educational level, poor compliance with recommended vaccinations in the past, and no reports of specified chronic conditions [29]. The simple aggregated results were similar to previous studies, but in our logistic regression analysis adjusting for multiple factors, we found no significant relationship with chronic conditions, nor did we find a significant relationship with sex or educational level.

With respect to information sources, a study from the United Kingdom and Ireland suggested that those resistant to a COVID-19 vaccination were less likely to obtain information about the pandemic from traditional and authoritative sources [30]. The results of our study also indicate that the vaccine-hesitant/refusal group is avoiding information from authoritative organizations because they receive their information from personal websites, SNS, and video sites. Information about the COVID-19 vaccine on private websites is significantly less reliable [31,32], and even some physicians disseminate information on their personal websites that is not based on scientific evidence [33]. Betsch et al. and Nan et al. have demonstrated that exposure to vaccine-critical websites and blogs negatively impacts the intention to vaccinate [34,35]. For example, during the COVID-19 pandemic, telecommunication masts across Europe, North America, and Australasia have been damaged or destroyed in arson attacks, while engineers have been subjected to verbal and physical abuse because of the misinformation on social networking sites that COVID-19 was propagating from telecommunication masts [36]. The public may also seek out and enable the spread of misinformation, which is rapidly distributed via social media [31]. On the Internet video site TikTok, only 36 videos encouraged receiving the COVID-19 vaccine; these videos garnered over 50% of the total cumulative views and just under 50% of the total likes. The 38 videos that discouraged the vaccine garnered 39.6% of the total cumulative views, 44.3% of the likes, and 47.4% of the comments [37]. This suggests that misinformation is more likely to spread because people are more interested in it [33].

Of course, information released by governments and public institutions is not always correct, but there is no doubt that it is at least more scientifically based than the information disseminated by individuals. Previous studies have shown that many vaccine refusers do not trust such public institutions [30]. Therefore, a large part of how one determines that information is incorrect depends on one’s personal beliefs. In Japan, a COVID-19 information website, Covi-Navi, has been established by infectious disease researchers and is independent of the government. In the present study, we also asked about the use of this site, but the percentage of use was low in both the vaccination and refusal groups. Although the site was reliable and well-known among researchers, it appears to not have been widely used by the general public. We believe that there is a need to reconsider the means of effectively communicating correct information.

On the other hand, the odds ratio of using TV as a source of information was significantly higher among those willing to receive the COVID-19 vaccine or who were already vaccinated. One possible reason for this is that the population in the COVID-19 vaccine willingness/already vaccinated group is relatively older in age. They are generally less exposed to current sources of information such as Internet video sites. In addition, the TV information about the COVID-19 vaccine was more impartial than it has been for other vaccines. In the past, many Japanese news programs have sensationalized adverse vaccine reactions to an excessive degree [38,39,40]. In light of this negative campaign, we found that this COVID-19 vaccine news coverage was based on the facts and was as dispassionate as possible [7].

There are some limitations to our study. First, this study was Internet-based; hence, we could not eliminate the selectivity bias. Second, this study was a cross-sectional study, and therefore, no causality could be established. In addition, the participants may have been affected by the available information on COVID-19 at the time of the survey (first week of December 2021). The questionnaire was designed to be simple; thus, we could not evaluate other socio-demographic factors such as details of residence, lifestyles, and habits.

## 5. Conclusions

Despite some limitations, our study confirmed that 80% of the general Japanese population has been vaccinated or is willing to receive the COVID-19 vaccine, and 20% is still refusing or is hesitant. As for sources of information related to vaccine refusal or hesitancy, this study found that it was less likely that respondents received information from television and newspapers but were more likely to receive information from personal websites, summary websites, and video sites such as YouTube and TikTok compared to vaccine willing or already vaccinated respondents. Although there is misinformation from social networking sites and websites, there is also correct information; if people can make good use of social networking sites and the media, they can obtain correct information about vaccination [41]. However, one implication is that it is very important to know how to eliminate misinformation and obtain correct information as a criterion for decision-making. The cause of vaccine refusal and hesitation is mixed with anxiety and disbelief due to misinformation. Therefore, it is important to dispel misinformation and help people make vaccination decisions by sharing correct information.

## Figures and Tables

**Figure 1 vaccines-10-01041-f001:**
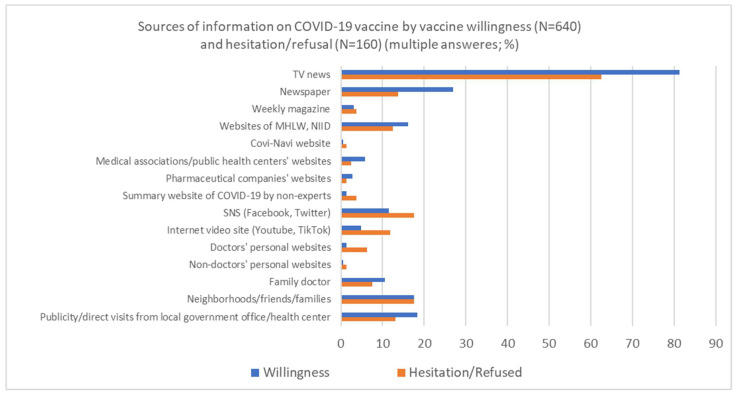
Sources of information on COVID-19 vaccine by vaccine willingness/hesitancy and refusal.

**Table 1 vaccines-10-01041-t001:** Characteristics of respondents.

		N	%
Sex	Males	418	52.2
	Females	382	47.8
Age group	Under 19 years	97	12.1
	20–29 years	156	19.5
	30–39 years	112	14.0
	40–49 years	119	14.9
	50–59 years	99	12.4
	60–69 years	99	12.4
	Over 70 years	118	14.7
Chronic diseases	None	598	74.8
	One or more	202	25.2
Education	Junior high school	48	6.0
	Senior high school	259	32.4
	Vocational school/college	108	13.5
	University	352	44.0
	Graduate school	33	4.1
Occupation	Self-employed	40	5.0
	Company employee	270	33.7
	Civil servant	26	3.2
	Medical expert	16	2.0
	Part-time worker	106	13.2
	Housekeeper	76	9.5
	Student	110	13.8
	Unemployed	142	17.8
	Others	14	1.8
Annual income	Under JPY 3,000,000	450	56.3
	JPY 3,000,000–6,000,000	216	27.0
	JPY 6,000,000–9,000,000	71	8.9
	JPY 9,000,000–12,000,000	33	4.1
	JPY 12,000,000–15,000,000	11	1.4
	Over JPY 15,000,000	19	2.3

**Table 2 vaccines-10-01041-t002:** Willingness to be vaccinated against COVID-19 by characteristics.

		Yes (%)	Unsure (%)	No (%)	*p* *
Sex	Males	349 (83.5)	35 (8.4)	34 (8.1)	0.033
	Females	291 (76.2)	49 (12.8)	42 (11.0)	
Age group	Under 19 years	68 (70.1)	18 (18.6)	11 (11.3)	<0.001
	20–29 years	101 (64.7)	37 (23.8)	18 (11.5)	
	30–39 years	84 (75.0)	8 (7.1)	80 (17.9)	
	40–49 years	104 (87.4)	8 (6.7)	7 (5.9)	
	50–59 years	86 (86.9)	5 (5.0)	8 (8.1)	
	60–69 years	87 (87.9)	4 (4.0)	8 (8.1)	
	Over 70 years	110 (93.2)	4 (3.4)	4 (3.4)	
Chronic diseases	None	461 (77.1)	72 (12.0)	65 (10.9)	0.002
	One or more	179 (88.6)	12 (5.9)	11 (5.5)	
Education	Junior high school	29 (60.4)	12 (25.0)	7 (14.6)	<0.001
	Senior high school	215 (83.0)	23 (8.9)	21 (8.1)	
	Vocational school/college	90 (83.3)	4 (3.7)	14 (13.0)	
	University	280 (79.5)	38 (10.8)	34 (9.7)	
	Graduate school	26 (78.8)	7 (21.2)	0 (0.0)	
Occupation	Self-employed	32 (80.0)	6 (15.0)	2 (5.0)	0.017 **
	Company employee	204 (75.6)	40 (14.8)	26 (9.6)	
	Civil servant	26 (100.0)	0 (0.0)	0 (0.0)	
	Medical expert	15 (93.8)	1 (6.2)	0 (0.0)	
	Part-time worker	86 (81.1)	3 (2.8)	17 (16.1)	
	Housekeeper	65 (85.5)	4 (5.3)	7 (9.2)	
	Student	85 (77.3)	13 (11.8)	12 (10.9)	
	Unemployed	115 (81.0)	16 (11.3)	11 (7.7)	
	Others	12 (85.8)	1 (7.1)	1 (7.1)	
Annual income	Under JPY 3,000,000	365 (81.1)	34 (7.6)	51 (11.3)	0.067 **
	JPY 3,000,000–6,000,000	168 (77.8)	31 (14.3)	17 (7.9)	
	JPY 6,000,000–9,000,000	59 (83.1)	8 (11.3)	4 (5.6)	
	JPY 9,000,000–12,000,000	27 (81.8)	4 (12.1)	2 (6.1)	
	JPY 12,000,000–15,000,000	8 (72.7)	3 (27.3)	0 (0.0)	
	Over JPY 15,000,000	13 (68.4)	4 (21.1)	2 (10.5)	

*: Pearson’s chi-square test. **: Possible problem with chi-square, as 20% of the cells have an expected frequency of less than 5.

**Table 3 vaccines-10-01041-t003:** Adjusted odds ratio (AOR) and 95% confidence interval (CI) of logistic regression analysis for willingness to receive COVID-19 vaccine.

Variables		Model 1		Model 2 †		Model 3 ‡	
		AOR	95%CI	AOR	95%CI	AOR	95%CI
Information sources	TV news	2.56 *	1.68–3.89	2.31 *	1.48–3.63	2.44 *	1.54–3.85
	Newspapers	1.86 *	1.10–3.13	1.29	0.74–2.25	1.23	0.69–2.19
	Weekly magazines	0.77	0.27–2.18	0.82	0.28–2.41	0.85	0.28–2.54
	Websites of MHLW/NIID	1.55	0.87–2.77	1.57	0.86–2.84	1.63	0.87–3.03
	Covi-Navi website	0.39	0.05–2.82	0.51	0.07–3.77	0.35	0.04–2.72
	Medical associations’/public health centers’ websites	1.92	0.58–6.27	2.04	0.59–6.99	2.51	0.70–8.94
	Pharmaceutical companies’ websites	3.01	0.48–18.56	3.83	0.56–26.0	5.21	0.68–39.60
	Summary websites of COVID-19 by non-experts	0.29	0.08–1.00	0.27 *	0.07–0.97	0.21 *	0.06–0.77
	SNS (Facebook, Twitter)	0.93	0.54–1.59	1.35	0.76–2.40	1.43	0.79–2.59
	Internet vide sites (YouTube, TikTok)	0.45 *	0.22–0.92	0.37 *	0.17–0.79	0.33 *	0.15–0.73
	Doctors’ personal websites	0.19 *	0.06–0.64	0.17 *	0.04–0.60	0.16 *	0.04–0.59
	Non-doctors’ personal websites	0.97	0.09–9.79	0.99	0.09–10.36	0.78	0.07–8.53
	Family doctor	1.44	0.70–2.95	1.22	0.56–2.64	1.18	0.54–2.59
	Neighborhoods/friends/families	1.02	0.62–1.70	1.16	0.69–1.97	1.09	0.63–1.86
	Publicity/direct visits from local government office/health center	1.69	0.97–2.88	1.32	0.74–2.36	1.47	0.80–2.69
Sex	Males			1	-	1	-
	Females			0.82	0.54–1.23	0.70	0.45–1.09
Age group	Under 19 years			1	-	1	-
	20–29 years			0.80	0.45–1.44	0.74	0.38–1.47
	30–39 years			1.23	0.64–2.35	1.03	0.49–2.16
	40–49 years			2.81 *	1.34–5.89	2.51 *	1.10–5.73
	50–59 years			2.46 *	1.12–5.42	1.98	0.82–4.77
	60–69 years			2.58 *	1.11–5.97	1.97	0.78–4.97
	Over 70 years			4.57 *	1.82–11.49	3.24 *	1.17–8.96
Presence of chronic diseases	None					1	-
	One or more					1.55	0.84–2.85
Education	Junior high school					1	-
	Senior high school					3.32 *	1.54–7.14
	Vocational school/College					3.14 *	1.23–8.00
	University					2.51 *	1.13–5.55
	Graduate school					1.69	0.50–5.68
Annual income	Under JPY 3,000,000					1	-
	JPY 3,000,000–6,000,000					0.62 *	0.38–0.99
	JPY 6,000,000–9,000,000					0.70	0.32–1.50
	JPY 9,000,000–12,000,000					0.76	0.26–2.21
	JPY 12,000,000–15,000,000					0.41	0.08–2.16
	Over JPY 15,000,000					0.42	0.13–1.35

*: *p* < 0.05. †: Model 2 was adjusted for sex and age group. ‡: Model 3 was adjusted for sex, age group, presence of chronic diseases, educational background, and annual income. Model evaluation: Model 1: Akaike’s information criterion corrected (AICc) 768.57, R^2^ 0.08; Model 2: AICc 745.18, R^2^ 0.13; Model 3: AICc 747.42, R^2^ 0.15. AOR: Adjusted odds ratio, CI: confidence interval, TV: television, MHLW: Ministry of Health, Labour and Welfare, NIID: National Institute of Infectious Diseases, SNS: Social Networking Service.

## Data Availability

Data presented in this study are available upon request from the corresponding authors (T.Y.). Data are not publicly available due to privacy concerns.

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
