# Peer review of "The Relationship between Sources of COVID-19 Vaccine Information and Willingness to Be Vaccinated: An Internet-Based Cross-Sectional Study in Japan"

_vaccines, 2022, doi:10.3390/vaccines10071041_

Round 1

Reviewer 1 Report

The paper analyses attitudes towards vaccination against Covid-19 in Japan. It compares people who were vaccinated or willing to be vaccinated to people who are hesitant (because of concerns about efficacy and safety) or those who refuse any vaccination.

The empirical analysis is based on 800 persons interviewed by internet in December 2021, about 10 months after the beginning of vaccination. Factors of vaccine hesitancy or refusal focus on demographic and socio-economic factors and on sources of information about vaccines (15 sources of information).

The paper is clearly written, well documented, the statistical methods are straightforward, results are sound and new, the discussion is appropriate.

Comments

1. The paper is short, and presents mainly the results of the statistical analysis. It could be somewhat improved by making more use of the social stratification presented in text. Two groups are indeed more reluctant than others:

- young people (below age 30), who are also more likely to have a lower income, to be in high school, to use social networks, to have fewer diseases, etc. (33% hesitant or refusing)

- very wealthy people, income > 12 M Yen (30% hesitant or refusing)

Authors could add a short paragraph (possibly a table) explaining how those two groups differ from the rest of the population, and add how they contribute to the non-acceptors in the population.

2. A comment on why young women are more reluctant/refusing than young men would be welcome, since in most countries women are more concerned with their health.

3. A comment on the role of information by television would be welcome: are those older persons? more educated, more concerned, better informed?

4. In the discussion, final paragraph (line 229 sq): information is not the sole criteria for making a decision. In democratic societies people have the right to refuse vaccination, for a variety of reasons (fear of side effects, religious, not feeling concerned, etc.).

Typos

Line 206: put [25] in brackets.

Line 225: “compare” (not compere)

Table 3: why adding “data” in Model 1 column?

Author Response

Dear Reviewer 1,

Thank you very much for your very useful comments.

In accordance with your comment, we have rewritten it as follows;

  1. The paper is short, and presents mainly the results of the statistical analysis. It could be somewhat improved by making more use of the social stratification presented in text. Two groups are indeed more reluctant than others:

- young people (below age 30), who are also more likely to have a lower income, to be in high school, to use social networks, to have fewer diseases, etc. (33% hesitant or refusing)

- very wealthy people, income > 12 M Yen (30% hesitant or refusing)

Authors could add a short paragraph (possibly a table) explaining how those two groups differ from the rest of the population, and add how they contribute to the non-acceptors in the population.

Answer 1. Thank you for your suggestion. We added explanations for above two groups in the discussion section (lines 198-210).

  1. A comment on why young women are more reluctant/refusing than young men would be welcome, since in most countries women are more concerned with their health.

Answer 2. Thank you for your opinion. We added some comments in the discussion section (lines 201-205)

  1. A comment on the role of information by television would be welcome: are those older persons? more educated, more concerned, better informed?

Answer 3. Thank you for your suggestion. We added some comments for the role of TV information in the discussion section (lines 271-279)

  1. In the discussion, final paragraph (line 229 sq): information is not the sole criteria for making a decision. In democratic societies people have the right to refuse vaccination, for a variety of reasons (fear of side effects, religious, not feeling concerned, etc.).

Answer 4. Thank you for your opinion. As you said, people also have the right to refuse vaccines for various reasons and that too must be respected. I changed the first version of our conclusion to the following sentences: "The cause of vaccine refusal and hesitation is mixed with anxiety and disbelief due to misinformation. Therefore, it is important to dispel misinformation and help people make vaccination decisions by sharing correct information".

Typos

Line 206: put [25] in brackets.

Line 225: “compare” (not compere)

Table 3: why adding “data” in Model 1 column?

Answers. Thank you for your comment. We have corrected all the errors you pointed out.

Reviewer 2 Report

In the manuscript by Yoda et al., the authors present results from a survey regarding the COVID-19 vaccine.  The manuscript is very well-written, easy to follow, and the results are clearly presented.  The authors also nicely state the limitations of their study.  I only have a few minor comments for improvement:

Line 169 - "corona vaccine" is colloquial.  Use COVID-19 vaccine or SARS-CoV-2 vaccine.

Lines 176-177 - call out the table and line to which these results are shown.

Line 198 - why is TikTok in quotes?

Author Response

Dear Reviewer 2,

Thank you very much for your very useful suggestions and feedback.

In accordance with your comment, we have rewritten it as follows;

Line 169 - "corona vaccine" is colloquial.  Use COVID-19 vaccine or SARS-CoV-2 vaccine.

 Answer. We changed the word from colloquial term to the technical term.

Lines 176-177 - call out the table and line to which these results are shown.

 Answer. We added the appropriate table and models where the results are shown (lines 211-217).

Line 198 - why is TikTok in quotes?

 Answer, This is simple mistake. We deleted quotes.